# Emergence to dominance: Estimating time to dominance of SARS-CoV-2 variants using nonlinear statistical models

Srishti Awasthi[1], Maryam Zolfaghari Dehkharghani[1], Miguel Fudolig[2]*

1 Department of Healthcare Administration and Policy, School of Public Health, University of Nevada Las Vegas, Las Vegas, Nevada, United States of America, 2 Department of Epidemiology and Biostatistics, School of Public Health, University of Nevada Las Vegas, Las Vegas, Nevada, United States of America

☯ These authors contributed equally to this work.

* miguel.fudolig@unlv.edu

**Data availability statement:** All data is available in the CDC data website: Phttps://data.cdc.gov/Laboratory-Surveillance/SARS-CoV-2-Variant-Proportions/jr58-6ysp/about_data. It is also attached as a ZIP file in the submission. The R Code is uploaded as an HTML file with the submission.

**Funding:** The author(s) received no specific funding for this work.

**Competing interests:** The authors have declared that no competing interests exist.

## Abstract

**Background/Objective**: Relative proportion of cases in a multi-strain pandemic like the COVID-19 pandemic provides insight on how fast a newly emergent variant dominates the infected population. However, the behavior of relative proportion of emerging variants is an understudied field. We investigated the emerging behavior of dominant COVID-19 variants using nonlinear statistical methods and calculated the time to dominance of each variant.

**Method**: We used a phenomenological approach to model national- and regional-level variant share data from the national genomic surveillance system provided by the Centers for Disease Control and Prevention to determine the best model to describe the emergence of two recent dominant variants of the SARS-CoV-2 virus: XBB.1.5 and JN.1. The proportions were modeled using logistic, Weibull, and generalized additive models. Model performance was evaluated using the Akaike Information Criteria (AIC) and the root mean square error (RMSE).

**Findings**: The Weibull model performed the worst out of all three approaches. The generalized additive model approach slightly outperformed the logistic model based on fit statistics, but lacked in interpretability compared to the logistic model. These models were then used to estimate the time elapsed from emergence to dominance in the infected population, denoted by the time to dominance (TTD). All three models yielded similar TTD estimates. The XBB.1.5 variant was found to dominate the population faster compared to the JN.1 variant, especially in HHS Region 2 (New York) where the XBB.1.5 was believed to emerge. This research expounds on how emerging viral strains transition to dominance, informing public health interventions against future emergent COVID-19 variants and other infectious diseases.

## Introduction

The COVID-19 pandemic, caused by the SARS-CoV-2 virus, a member of the coronavirus family, has had a significant impact on global health, economy, and societies. Coronaviruses,

including COVID-19, Middle East Respiratory Syndrome (MERS), and Severe Acute Respiratory Syndrome (SARS), are a group of respiratory viruses that can cause diseases in animals and humans. They are named after the crown-like spikes on their surface. Human coronaviruses were first discovered in the mid-1960s, and public health officials closely monitor them. The SARS-CoV-2 virus, prone to genetic alterations over time, experiences mutations during replication. Since the emergence of the SARS-CoV-2 virus, it has undergone dynamic evolution characterized by various changes resulting in the emergence of different strains. The ongoing mutation of SARS-CoV-2, a characteristic of RNA viruses, occurs due to errors in viral replication, as RNA polymerases lack proofreading mechanisms to ensure high fidelity [1]. These genetic changes lead to the emergence of new variants, some of which have significant public health implications due to increased transmissibility or severity of illness [2]. These mutations can cause an emergence of new variants that have different characteristics such as vaccine resistance, immunity resistance and reinfection. The genetic heterogeneity of the virus enables it to adapt and perhaps improve its ability to survive and transmit. In order to address this issue, it is imperative that we closely monitor these mutations and adjust our interventions accordingly.

## Recently emerged COVID-19 variants

Since the onset of the pandemic, several variants of SARS-CoV-2 have been identified and classified by health authorities as Variants of Interest (VOIs) or Variants of Concern (VOCs). Notable variants include Alpha (B.1.1.7), first detected in the United Kingdom; Beta (B.1.351), first detected in South Africa; Delta (B.1.617.2), first detected in India; and Omicron (B.1.1.529), first detected in Botswana and South Africa; with each variant showing unique traits and health impacts [2,3].

The Omicron variant (B.1.1.529) emerged in late 2021 and was first detected in Botswana and South Africa and quickly spread all over the world. The Omicron variant of the SARS-COV-2 virus was notable for its numerous spike protein mutations, raising concerns about immune evasion and vaccine effectiveness [4,5]. Two of the most recent variants that dominated the infected population in the United States (US) are XBB.1.5, also known as the Kraken variant, and JN.1. The XBB.1.5 variant is a descendent lineage of XBB, which is a recombinant of Omicron BA.2.10 and BA.2.75 descendant lineages [6–8]. The XBB.1.5 emerged in New York City and rapidly spread in the region in November–December 2022 and was identified to be responsible for most of the cases in the national level in early 2023. On the other hand, the JN.1 variant is the offspring of the Omicron BA.2.86 variant was first identified in the US in September 2023 and has already become the dominant variant of COVID-19 infections in the US. The mathematical modeling of the emergence and rapid spread of the XBB.1.5 and JN.1 variants remains a significant gap in the current research landscape. As of press time, there have not been any published research on the mathematical modeling of the JN.1 strain in the United States. Cheng et al. [9] used a multi-strain SIR model and variant proportion data to estimate the transmission rates and reproduction number of the XBB strain using surveillance data and variant proportions data. One of their recommendations was to include XBB.1.5 as a separate compartment as it dominated other XBB subvariants. However, there was no discussion on how quickly the XBB.1.5 dominated the XBB infections compared to other strains. Although there is a plethora of studies on the mathematical modeling of multi-strain epidemics, most of these studies are concerned with calculating transmissibility coefficients [10,11], estimating reproduction numbers [10,12,13], and performing stability analyses [14,15]. Current epidemic models often focus on well-established parameters,

neglecting a crucial element: the emergence and dominance of highly transmissible COVID-19 variants. Understanding how these new variants arise and outcompete existing variants is essential for accurate modeling in multi-strain epidemics. Furthermore, it is vital to analyze the behavior of these dominant variants within the US population to effectively predict future spread and inform public health interventions. The relative rate of emergence of each variant can be monitored using variant proportion data from genomic surveillance, which is provided by the Centers for Disease Control and Prevention (CDC) [16] and the Global Initiative on Sharing Avian Influenza Data [17], and wastewater surveillance [18,19]. An important parameter in describing the rate of dominance of a newly emerged variant is the time required for an emergent variant to reach the majority status in the infected population, also known as the time to dominance (TTD).

## Time to dominance

With the rapid mutation of SARS-CoV-2 in the United States general population, there is a critical need to determine how fast a newly emerged variant dominates the infected population. Newly emerged variants might exhibit vaccine resistance, increased transmissibility, and higher mortality rates. Once a newly emerged variant with these characteristics make up the majority of the infections, stricter public health policies might need to be reinstated to curb the spread of these variants. For instance, mask mandates were reinstated in some parts of the United States in early 2022 because of the emergence of the Omicron variant [20,21]. Public health officials have limited time to decide on the best strategy to characterize and devise strategies against these new emergent variants. One benchmark we can to measure the speed of emergence of a viral strain is the time to dominance (TTD). We define the TTD as the time it takes for an emergent viral strain to make up the overall majority, i.e. at least 50%, of the infected cases in a multi-strain epidemic. Fudolig [22] performed a simulation-based experiment to test the effect of vaccination and transmission on the TTD on a two-strain epidemic model. A three-parameter logistic growth model was used to describe the increase of variant share in the simulated epidemics and estimate TTD. Simulations showed that a more transmissible emergent strain relative to the existing strains was found to dominate the infected population faster. In addition, higher vaccination rates and coverages could lead to lower TTD. However, this method was not applied to data from real-world multi-strain epidemics such as COVID-19. While the use of the logistic model to estimate the TTD values for each simulation was sufficient for simulated data, other growth models such as a generalized logistic or Weibull growth models might yield a better fit and explain the emergence behavior better than logistic models. Semi-parametric approaches such as generalized additive models would also be a great option in modeling the non-linear growth of rapidly emerging SARS-CoV-2 variants. To the best of our knowledge, there has not been any prior research that compared the model performance of different models of COVID-19 variant proportion shares.

## Objectives and significance of the study

The JN.1 and XBB.1.5 variants were two of the most recently emerged variants to record variant shares of over 50% based on genomic surveillance data. This study estimated the duration required for the JN.1 and XBB.1.5 variants to dominate the COVID-19 infected population in the United States. Specifically, we focused on modeling the emergence of these variants using the following modeling approaches: the logistic growth model, the Weibull model, and the generalized additive model (GAM). These models were then used to estimate the TTD based

on variant proportion data reported by the CDC from 2021-2024 [16]. While the CDC provides model-based projected estimates of the variant share for the weeks when samples are being processed, the underlying data-driven model is not provided by the CDC. The results of this study presents preliminary data that would inform future studies on the emergence of new variants in multi-strain epidemic models. Only the weighted estimates of the variant share during reported dates were modeled in this study. The confidence limits provided by the CDC were not analyzed. By investigating the behavior of variant proportion shares of COVID-19, we addressed the literature gap on the behavior leading to the dominance of emergent variants in multi-variant epidemics.

## Materials and methods

### Dataset

COVID-19 variant proportion data was downloaded from the CDC website [16] on May 7, 2024. The variant proportions were calculated based on sequenced samples collected from different regions of the US. The regional division was defined by the United States Department of Health and Human Services (HHS). The member states of each region can be accessed through the following link: https://www.hhs.gov/about/agencies/iea/regional-offices/index.html. The dataset included variant proportions every two weeks from January 2022 to April 2024. We investigated the most recent SARS-CoV-2 variants, JN.1 and XBB.1.5, that were reported to have a majority of COVID-19 cases nationwide.

## Theory and calculation

### Variant proportions

Historically, there are multiple SARS-CoV-2 variants that coexist in different regions of the US. Fig 1 shows the estimated proportion share of the JN.1 variant as reported by the CDC. These estimates were recorded every two weeks based on empirical genomic sequencing data. The share of SARS-CoV-2 variant proportions appear to follow a non-linear trend in time. We modeled this non-linear trend using three different approaches: generalized additive models (GAM), logistic function curve fit, and Weibull function curve fit.

### Modeling non-linear emergence

**Generalized additive models.** Generalized additive models (GAM) are generalized linear models that utilize smoothing splines to accurately model non-linear trends [23]. GAMs are made up of a linear combination of a linear predictor and smoothing functions to describe any smooth monotonic curves, making it ideal to model variant proportion data. An intercept-only GAM was used in this study which had the following form:

$$f(t) = \beta_0 + s(t), \tag{1}$$

where $\beta_0$ is the model intercept and $s(t)$ is the smooth function of the time $t$. The semi-parametric nature of GAMs make it difficult to provide confidence intervals of inverse estimates such as the case of estimating TTD values, which will be explained in the section "Estimating Time to Dominance (TTD)". The `mgcv` package in R was used to fit a GAM on the CDC variant proportion share data.

**Logistic and Weibull models.** We fit non-linear functions such as the logistic and Weibull functions to model the non-linear increase of JN.1 and XBB.1.5 variant proportions

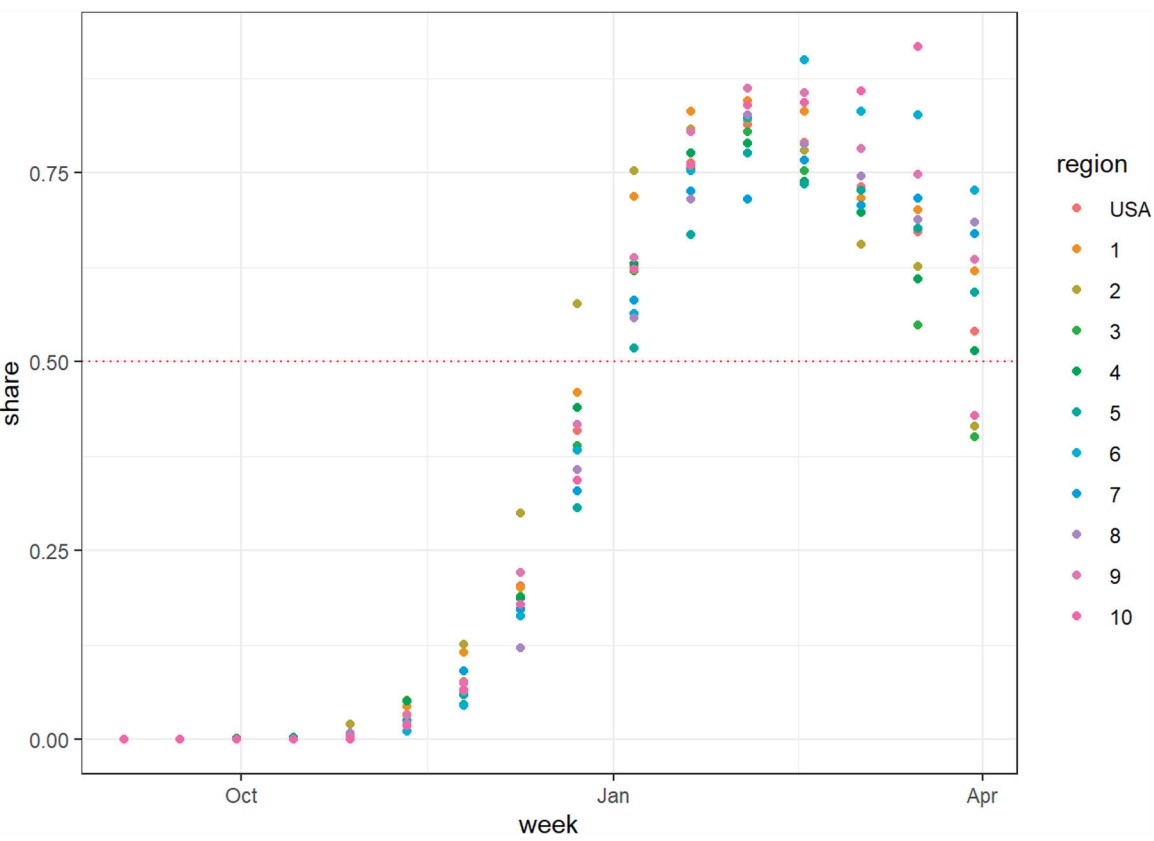

**Fig 1. National- and regional-level variant proportion data of the JN.1 subvariant.**

upon emergence. Fudolig [22] previously used the logistic function to model the increase in emerging variant proportion in a multi-strain epidemic in calculating TTD values. The five-parameter generalized logistic function can be expressed as

$$f(t) = c + \frac{d - c}{\left[1 + \exp(b(t - e))\right]^f},\tag{2}$$

where the time $t$ will be measured in days from the first emergence. $b$ is the scale factor, $c$ and $d$ are the respective upper and lower asymptotes of the logistic curve, $e$ is the inflection point, and $f$ is the asymmetry factor. The logistic function is also known as the Boltzmann sigmoidal function [24]. A closely related model to the logistic function is the Weibull function. The Weibull function is another function typically used to model growth curves that provides more flexibility in modeling non-monotonic functions. The Weibull function can be expressed as

$$f(t) = c + (d - c)(\exp[-\exp(b(\log(t) - e))]),\tag{3}$$

where $b$ is a scale factor, $c$ is the lower asymptote, $d$ is the upper asymptote, and $e$ is the point of inflection. Both models have been used to describe growth of the infected population in epidemics [25–29], but there is a literature gap in using these models to describe variant proportion shares. Curve fitting for both functions can be implemented using the R package `drc`, which is commonly used to model dose response curves [24]. Even though both logistic and

Weibull models are lacking in flexibility compared to the GAM, both logistic and Weibull models offer easily interpretable results that could translate to action items for policy makers and public health officials.

### Estimating the time to dominance (TTD, $t_D$).

The time $t = 0$ was set two weeks before a share greater than 0.01% was reported for the variant. Based on the criterion for $t = 0$ being the reporting date before the variant proportion share was reported to be above 0.01%, the start of the emergence occurred on October 15, 2022 for XBB.1.5 and Sept 2, 2023 for JN.1. All models were implemented from $t = 0$ to $t_{MAX}$, the time at which the variant proportion share is maximum for each region and variant. The fit of each model was assessed using the Akaike information criteria (AIC) and the root mean square error (RMSE). Lower AIC and RMSE values imply a better model fit on the data.

The values of $t_D$ were estimated for regional and national level data. The TTD value, denoted by $t_D$, were estimated for each model by numerically solving for $t_D$ such that $f(t = t_D) = 0.5$. The R packages `drc` and `mgcv` were used to implement the logistic, Weibull, and generalized additive models on the regional and national variant proportion estimates from the CDC. We wish to clarify that the $t_D$ does not equate to ED50, defined in the context of the study as the time required to reach 50% of the maximum share [30], as neither JN.1 and XBB.1.5 reached a maximum share equal to 1. The root finding function `uniroot` in R was used to estimated $t_D$ for both JN.1 and XBB.1.5. The TTD estimates obtained from each model were compared and analyzed for each region.

## Results and discussion

### Model performance

Figs 2 and 3 show the graph of the absolute values of the AIC and RMSE values for each model as applied to the national and regional COVID-19 variant proportion data. Lower AIC and RMSE values suggest a better model fit. The Weibull model appears to perform the worst out of the three models, while the logistic model and GAM performed comparably well in estimating the variant proportion curve during the emergence of the two variants.

The GAM outperformed both logistic and Weibull curve fits in modeling the emergence of both variants at the national level. Figs 4 and 5 show the three different models superimposed with the actual variant share data for XBB.1.5 and JN.1, respectively. The GAM and logistic model cannot be visually perceived implying very similar fit. Upon visual inspection, the Weibull model seems to underestimate the "knee" of the sigmoid curve of the XBB.1.5 variant share. The Weibull model fails to correctly account for the gradual increase of the variant share at $t = 70$ and $t = 84$, but appears to intersect the other two models close to the line $y = 0.5$. This behavior is reflected in Fig 3 which displays how the RMSE of the Weibull model is significantly higher compared to the other two models.

This large discrepancy in model performance is also observed in the regional level. The Weibull model performed relatively poorly in modeling the emergence of JN.1 and XBB.1.5 in all regions. Fig 6 displays the performance of the three models against the actual XBB.1.5 variant proportion data in HHS Region 5 (Chicago), which also shows the Weibull underestimation at $t = 70$ and $t = 84$ and the similarity between GAM and logistic models observed in the national data. The same trend was observed in the JN.1 variant proportion as shown in Fig 7, which includes the three models and the actual JN.1 variant share data in HHS Region 1 (Boston).

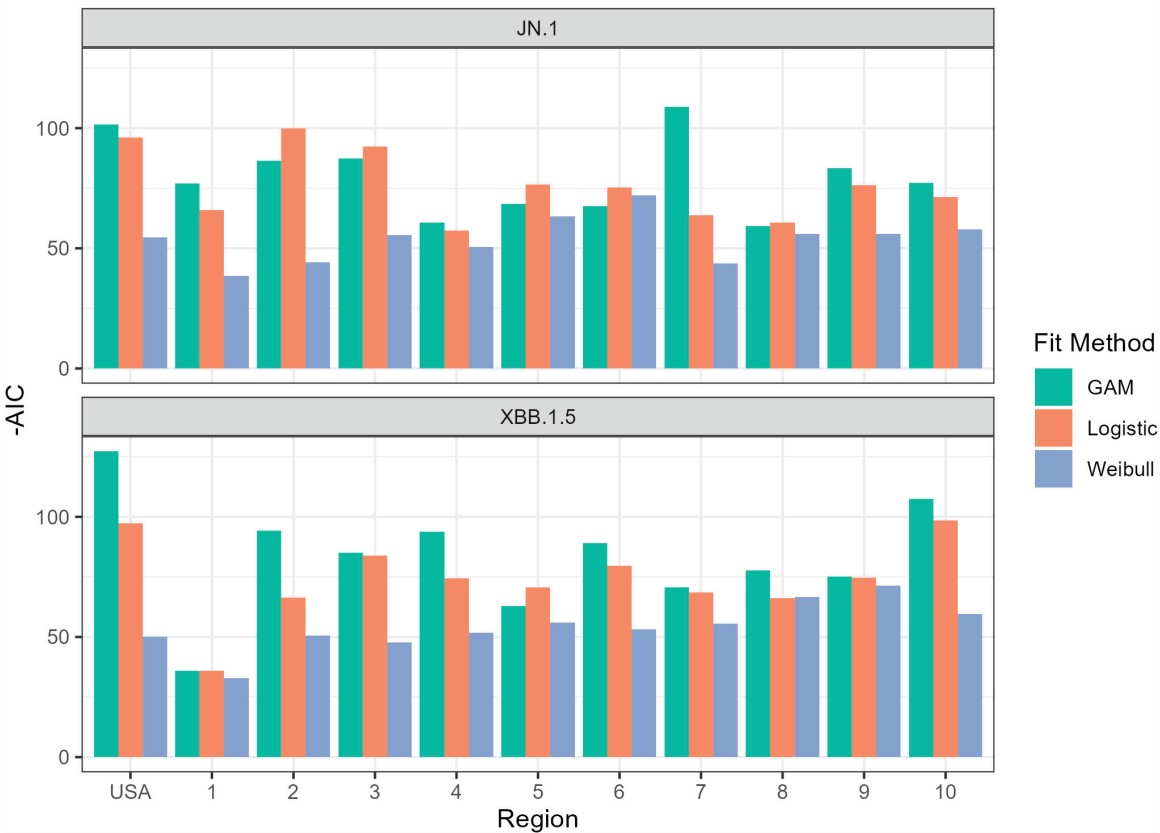

**Fig 2. The magnitude of the AIC values (-AIC) for each fit function plotted for the national and regional variant proportions data set**. All calculated AIC values were negative. A higher magnitude of the AIC corresponds to a better fit.

It is noteworthy that the Weibull model underestimation at the "knee", specifically $t = 70$ and $t = 84$, occur for both strains at the national level. For XBB.1.5, the dates corresponding to $t = 70$ and $t = 84$ are December 24, 2022 and January 7, 2023, respectively. During these weeks, festive holidays such as Christmas, Hanukkah, and New Year's Eve often entail numerous social gatherings and meetups that could have potentially lead to super-spreader events. After these dates, the variant proportion increased before plateauing at the maximum which was recorded in mid-March (March 18, 2023). As for JN.1, $t = 70$ and $t = 84$ occurred on November 11, 2023 and November 25, 2023. These weeks include Thanksgiving 2023, which was celebrated in the US on November 17, 2023. Like the aforementioned holidays, Thanksgiving is a holiday in the US that involves social and family gatherings all over the country. It is possible that the Weibull model could not account for the sudden increase in variant share leading up to reaching the simple majority of the variant proportion cases for both cases.

The poor performance of the Weibull model in this study contrasts with previous studies that assess the performance of the Weibull function in modeling COVID-19 incidence, prevalence, and mortality data. Attanayake et al. [31] determined phenomenologically that the Weibull growth curve performed the best in modeling the cumulative number of COVID-19 infections in the US from the case of first appearance to July 2, 2020. Al-Ani et al. [25] also found that the Weibull model performed the best in modeling the cumulative number of COVID-19 cases in Saudi Arabia. There is a plethora of research that use modified

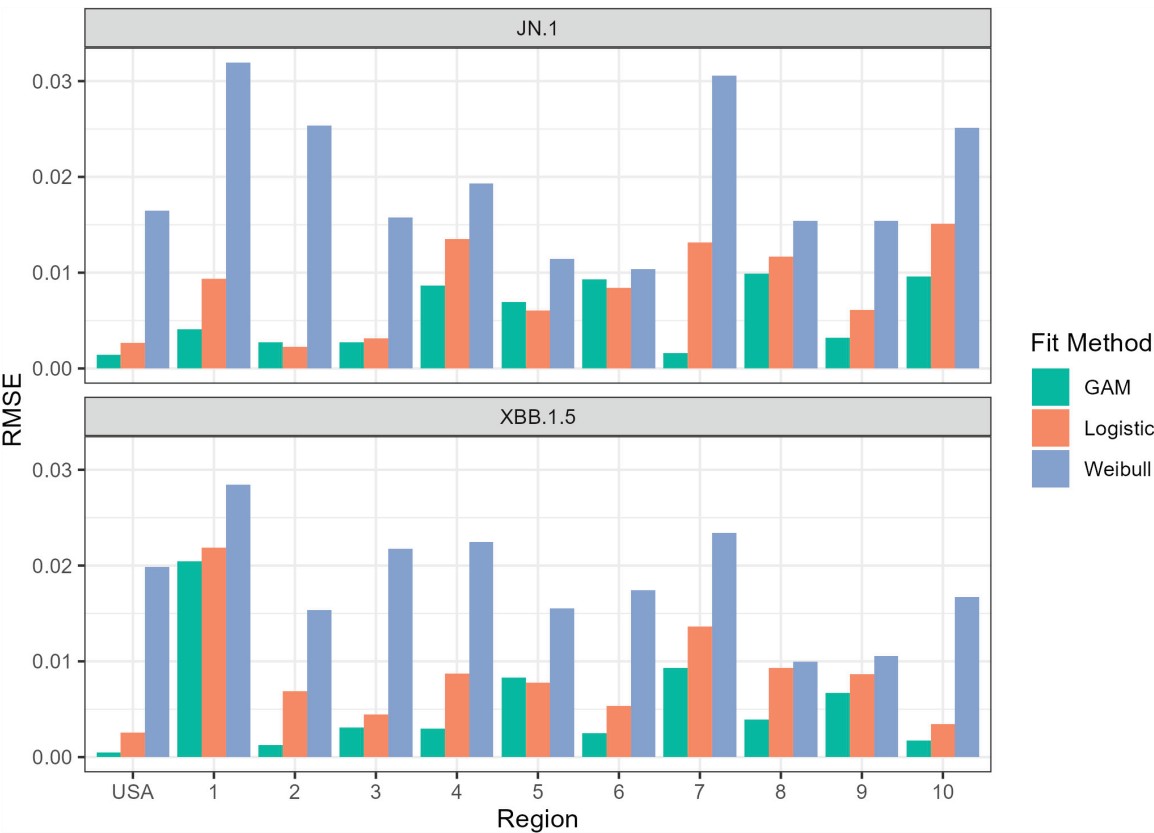

**Fig 3. The root mean square error (RMSE) values for each fit function plotted for the national and regional variant proportions data set.** A lower magnitude of the RMSE corresponds to a better fit.

Weibull distributions to model daily and cumulative number of cases and deaths of COVID-19 patients from all over the world [29,32,33]. Despite its wide usage in modeling other aspects of the COVID-19 pandemic, the Weibull function appears to perform underwhelmingly in modeling COVID-19 variant proportion share data in the United States. Based on our findings, the logistic or generalized additive models are more recommended to use in modeling the share of emerging COVID-19 variants that achieve majority dominance in the United States. While both logistic model and GAM perform similarly, the two models provide different insights about the curve. The logistic model provides more information on the shape of the curve, i.e. asymptotes, slopes, and inflection point. Moreover, it can provide an estimate of the share at any given time. Its limitations lie when we want to model the behavior of the variant proportion share past the emergence phase. After COVID-19 variants reach their maximum share, the variant share would naturally start decreasing as the number of cases decrease. This decrease can be naturally observed or could have been caused by the emergence of a more transmissible variant. The GAM approach would be more useful if a researcher intends to investigate the entire variant share curve. GAMs provides precise estimates for variant proportion shares during and after the emergence phase without constraint. While the GAM is generally the better performing model, the model does not have an interpretable closed form because of the general nature of the smoothing function $s(t)$. It is essential to establish specific research questions to determine which of the two models should be

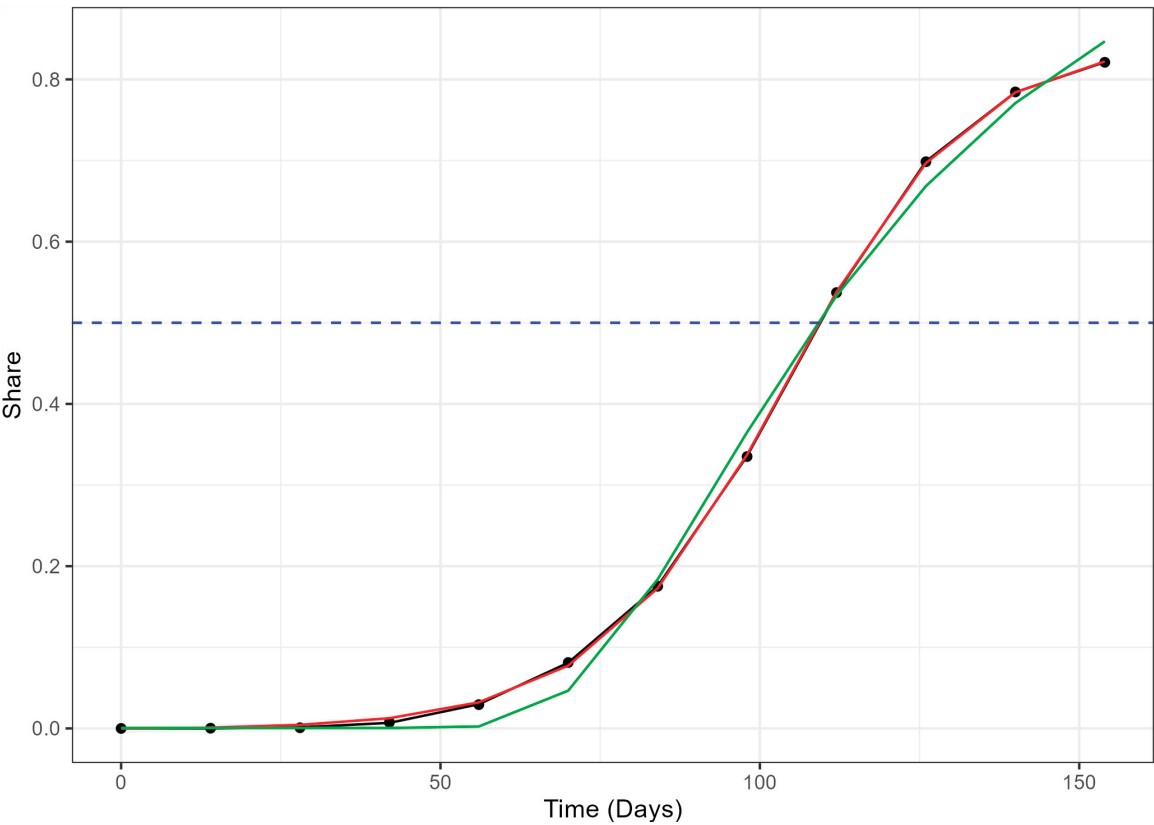

**Fig 4. The actual variant proportion data for XBB.1.5 in the US, shown as black dots, plotted with the GAM (black line), logistic model (red), and Weibull model (green) fitted values.** The blue dashed line marks the 50% share level used for calculating the TTD values.

used in analyzing variable proportion shares. In the next section, we illustrate how these models can be used to measure the time to dominance of the XBB.1.5 and JN.1 strains, which is an important aspect in characterizing a variant's emergent pattern.

## Model application: Estimating TTD ($t_D$)

Fig 8 shows the estimates of the $t_D$ values for the two variants for all models as applied to national and regional level data. While there are discrepancies between the models with respect to their AIC and RMSE values, the differences between $t_D$ estimates of the three models are quite low. The highest difference between the $t_D$ estimates from each model was 1.70 days for JN.1 (HHS Region 10, GAM vs. Weibull) and 2.04 days for XBB.1.5 (HHS Region 4, GAM vs. Weibull). In most cases, the GAM estimates are slightly higher compared to both logistic and Weibull estimates. Even though it did not perfectly capture the entire growth curve, the Weibull model's performance in estimating $t_D$ was promising for JN.1 and XBB.1.5. This observation was not surprising as Figs 4–7 showed the three curves converging close to $y = 0.5$ as mentioned in Model Performance section in the Results. For ease of reporting from this point onwards, we averaged the $t_D$ estimates from all three models.

Fig 9 shows the side-by-side comparison of the $t_D$ values for both JN.1 and XBB.1.5 variants in each region. The XBB.1.5 variant yielded lower time to dominance estimates across all regions except HHS Region 9 (San Francisco), in which the JN.1 variant had a slightly lower

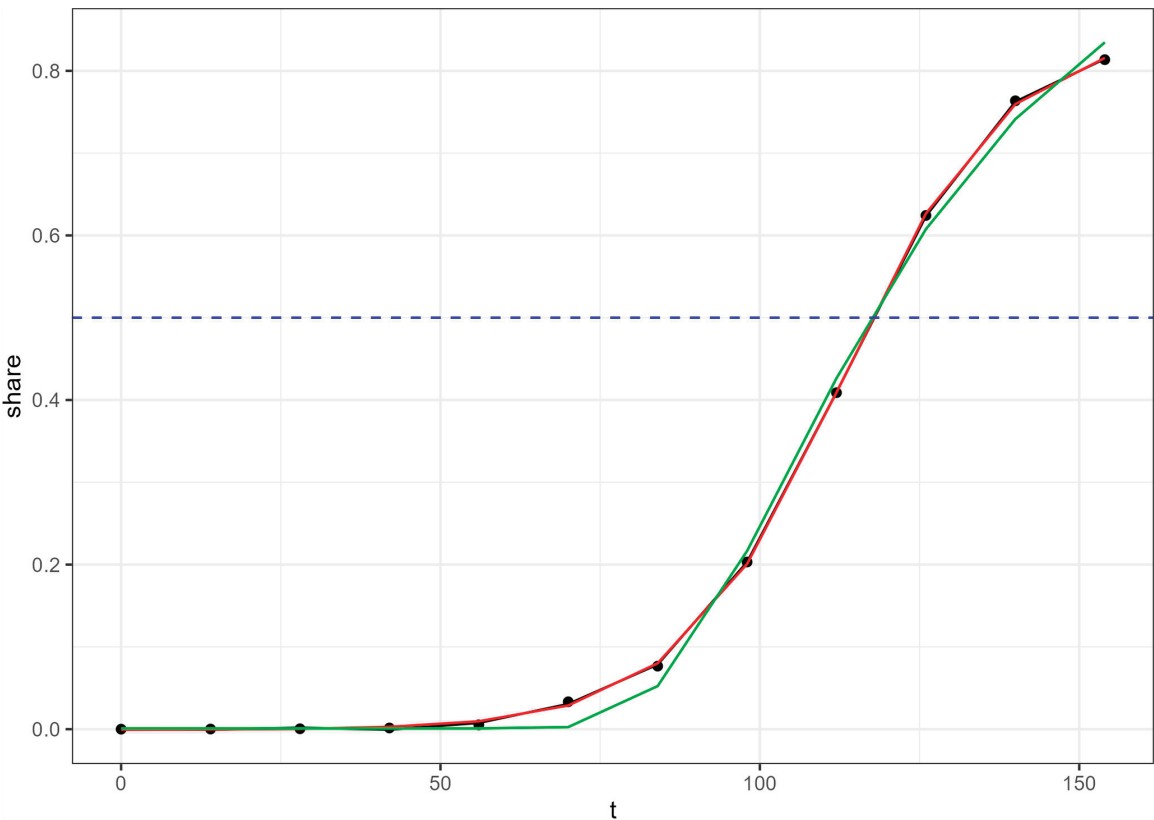

**Fig 5. The actual variant proportion data for JN.1 in the US, shown as black dots, plotted with the GAM (black line), logistic model (red), and Weibull model (green) fitted values.** The blue dashed line marks the 50% share level used for calculating the TTD values.

estimate. Even though the CDC insinuated that the JN.1 is more transmissible compared to other variants present in December 2023, the CDC did not find an increased risk to public health. Individuals who have updated vaccinations were also reported to be protected from the JN and XBB variants [34]. HHS Region 2 (New York) yielded the lowest $t_D$ value for both XBB.1.5 and JN.1 with respective $t_D$ estimates of 79.28 and 107.82 days. On the other hand, the highest $t_D$ GAM estimates for XBB.1.5 was recorded to be in HHS Region 10 (Seattle) at 124.73 days. It is important to highlight that the time to dominance for HHS Region 2 was typically the time where the "knee" of the logistic curve occurred. We can observe in Fig 10 that the "knee" appeared earlier compared to other regions shown in Figs 4–7. According to Luoma [6], New York could be the epicenter of the XBB.1.5 rapid spread which can explain the lower time to dominance in HHS Region 2 and surrounding HHS regions. HHS Regions 1 (Boston, $t_D$ = 88.58 days) and 3 (Philadelphia, $t_D$ = 95.97) closely follow as second and third lowest TTD values. On the other hand, HHS Region 10 (Seattle) is located in the opposite coast of the country and has a lower population density than HHS Regions 1, 2, and 3, which could have contributed to a slower spread of the variant. Meanwhile, the highest $t_D$ estimates for JN.1 was observed in HHS Region 5 at 125.63 days. The rapid dominance of JN.1 and XBB.1.5 variants in eastern HHS Regions (1, 2, and 3) likely stems from the extensive inter-connectivity of urban centers in these regions. Air, land, and water transport between these urban centers are possible, which could not be said for other HHS regions.

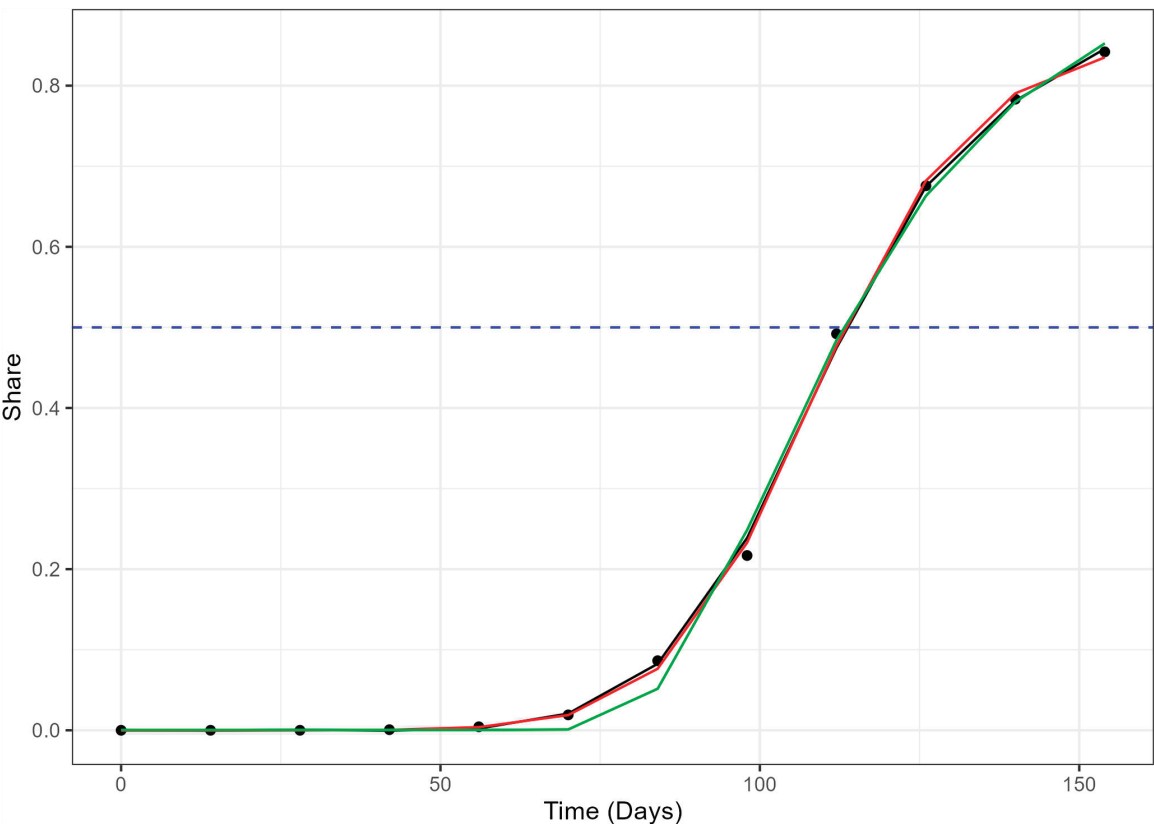

**Fig 6. The actual variant proportion data for XBB.1.5 in HHS Region 5, shown as black dots, plotted with the GAM (black line), logistic model (red), and Weibull model (green) fitted values.** The blue dashed line marks the 50% share level used for calculating the TTD values.

We must exercise caution on using these models for predicting the TTD values for an emergent variant as there are many factors that could affect the temporal trajectory of the variant proportion shares. A new VOI could be expected to dominate the infected population and its TTD could be predicted using models we used in the previous sections, but this expected dominance could be hindered by the emergence of another variant that could compete with the original VOI. Another limiting aspect of predicting TTD values of emerging variants is the time required to complete genomic sequencing of the samples. Adding intermediate data to predict the TTD of an emergent variant would suggest more frequent reports of genomic surveillance, but this might not be possible due to logistic constraints. Instead of predicting the TTD values, we can focus on estimating TTD values in simulation-based studies of multi-strain epidemics [22]. The emergence of other variants and temporal resolution of reports can be accounted for in the simulations, and the TTD for the emergent variant can be estimated using the models demonstrated in the preceding sections. The estimated TTD values from the simulations could be compared or tested for differences using other statistical methods.

## Limitations of the study

The phenomenological approach of the study provides insight on the temporal behavior of variant proportions for XBB.1.5 and JN.1. However, we must remain cautious of generalizing

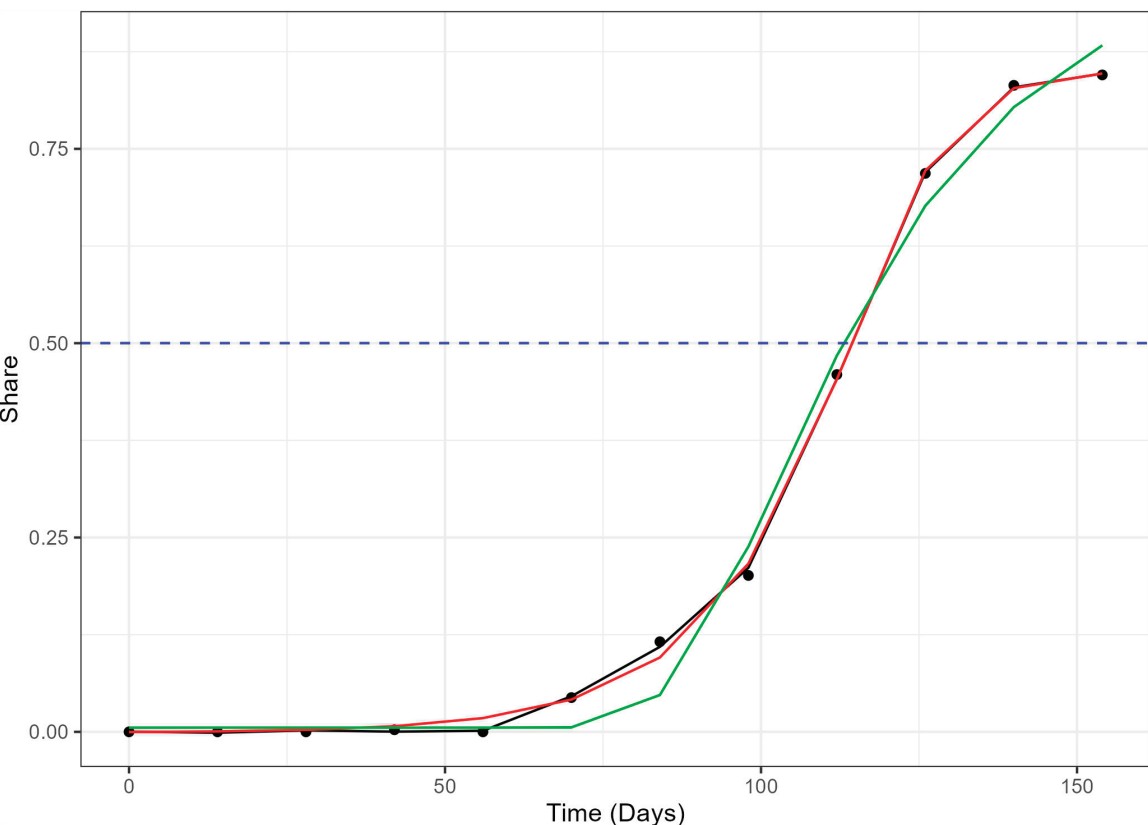

**Fig 7. The actual variant proportion data for JN.1 in HHS Region 1, shown as black dots, plotted with the GAM (black line), logistic model (red), and Weibull model (green) fitted values.** The blue dashed line marks the 50% share level used for calculating the TTD values.

this approach to previous dominant variants like the Alpha, Delta, and early Omicron variants because of the different public health interventions in place at the time of emergence. XBB.1.5 and JN.1 emerged after vaccines were provided to the public, mask mandates were dropped, and social distancing protocols were not enforced. We directed our study to the two most variants as they are the more relevant to current public health officials dealing with relaxed interventions. A separate study must be done to investigate the time to dominance for earlier variants.

We would also emphasize that these trends were analyzed from the genomic surveillance data of COVID-19 in the United States provided by the CDC. While this study focused on genomic sequencing data, future research will incorporate estimated variant proportions from wastewater surveillance. This rich data source holds promise for revealing potentially divergent trends in newly emerged COVID-19 variants.

## Conclusion

We have modeled the variant share data during the emergence of two variants that dominated the COVID-19 infected population in the US most recently: XBB.1.5 and JN.1. Logistic, Weibull, and generalized additive models (GAMs) were considered for both national and regional-level data for the proportion of confirmed XBB.1.5 and JN.1 cases provided by the CDC. After evaluating model performance based on AIC and RMSE values, the Weibull

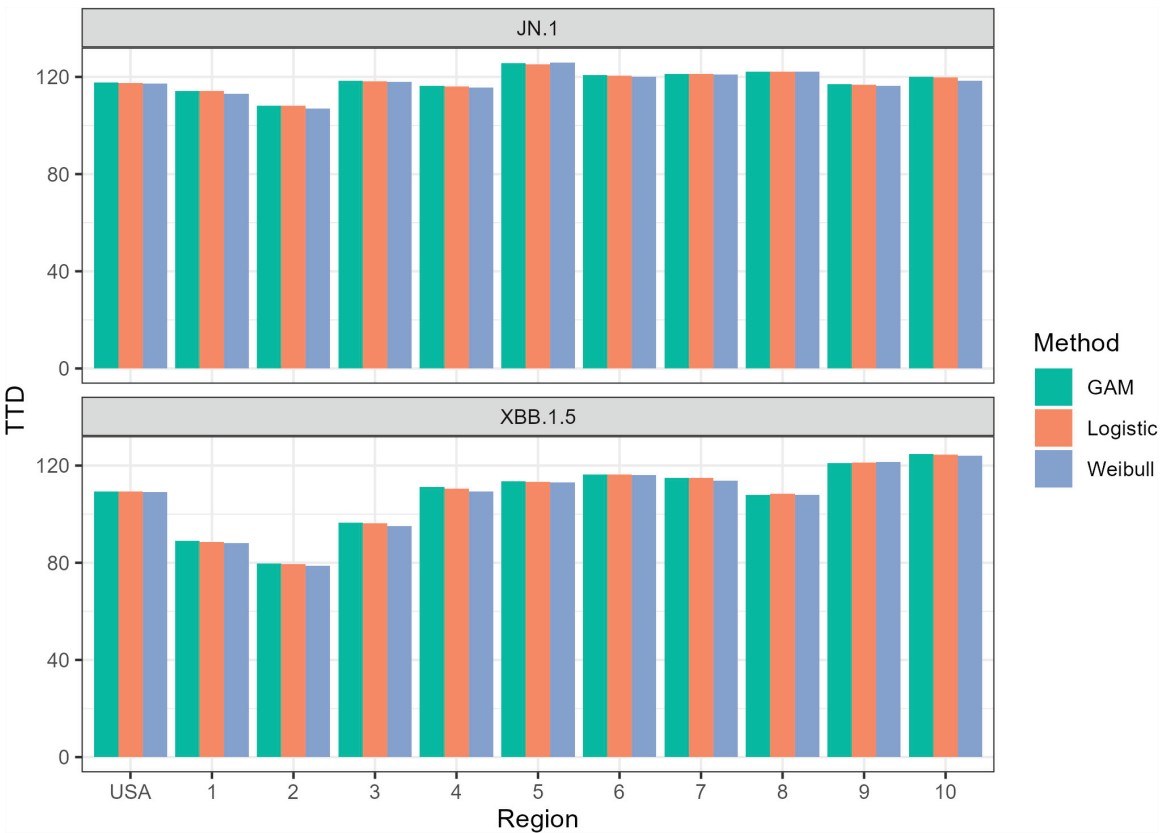

**Fig 8. National- and regional-level estimates of the $t_D$ values for JN.1 and XBB.1.5.**

model was determined to perform the worst among the three models. The logistic and GAM approaches yielded similar results, with GAM providing slightly lower RMSE values. The advantage in model performance and versatility provided by GAM could be compensated with the interpretability of the logistic model in modeling variant emergence in a multi-variant epidemic such as COVID-19.

We were also able to calculate the time to dominance (TTD) $t_D$, which measures the time required for the variants to reach majority status in the infected population. Despite its sub-par performance in modeling variant emergence, the $t_D$ estimates from the Weibull model did not deviate largely from the $t_D$ estimates from the logistic model and GAM. We also determined that the TTD was the lowest in HHS Region 2 (New York), which indicates a faster spread of XBB.1.5 and JN.1 during emergence compared to other regions. The dominance of JN.1 and XBB.1.5 variants concentrated in eastern HHS Regions (1, 2, and 3) likely stems from these regions' unique characteristics. These regions boast densely populated urban centers extensively interconnected by air, land, and water transportation networks, facilitating rapid viral spread compared to other regions, where public transport linking urban centers is limited. The TTD estimates for both variants ranged from 2.6 to 4 months depending on the region, which could be used as an early temporal checkpoint to assess whether current strategies against COVID-19 infections should change to fight against these new variants.

We also observed how the timing of the "knee" of the variant share curve could affect whether new variants could dominate the population. The rapid increase of both XBB.1.5

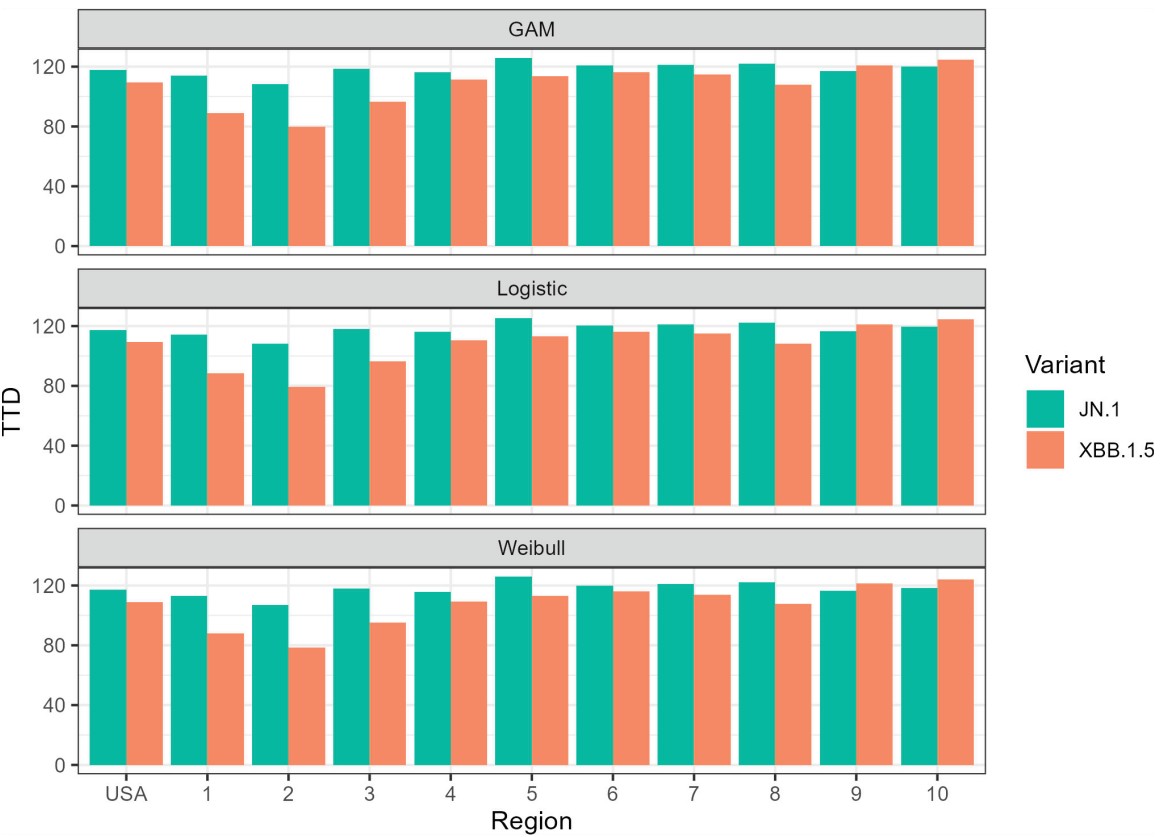

**Fig 9. Side-by-side comparison of $t_D$ values for JN.1 and XBB.1.5 evaluated for each model.**

and JN.1 variant shares occurred during holiday seasons in which people in the US typically gather in large groups, which might have made it easier for these variants to dominate. This temporal association suggests that holiday gatherings might act as a catalyst in decreasing the TTD of new variants in the US. We could use the findings of this study to analyze the similarities in the emergence of the FLiRT variants (KP.2, KP.3, KP.1.1) compared to the previously dominant variants such as JN.1 and XBB.1.5 in this period where COVID-19 policies is less stringent compared to the policies during the pandemic. While none of these FLiRT variants have a 50% share of the cases as of press time, the results of the study could inform which one of these variants could dominate the infected population in the US in the following weeks, especially after the independence day weekend celebrations.

In addition to researching future emerging COVID-19 variants, further research is needed to determine relative model performance between logistic, Weibull, and GAM approaches hold for other variants of interest (VOIs) such as BA.2.86 that achieved majority status before the relaxation of COVID-19 public health and social measures. We would also like to include VOIs that did not achieve majority status such as EG.5 and HV.1 in future work. Expanding the study on data from outside the United States is also recommended to enable us to compare different optimal control strategies in containing the spread of newly emergent COVID-19 variants all around the world. Exploring avenues for TTD prediction after incorporating the models presented and simulation-based studies is also recommended for further study.

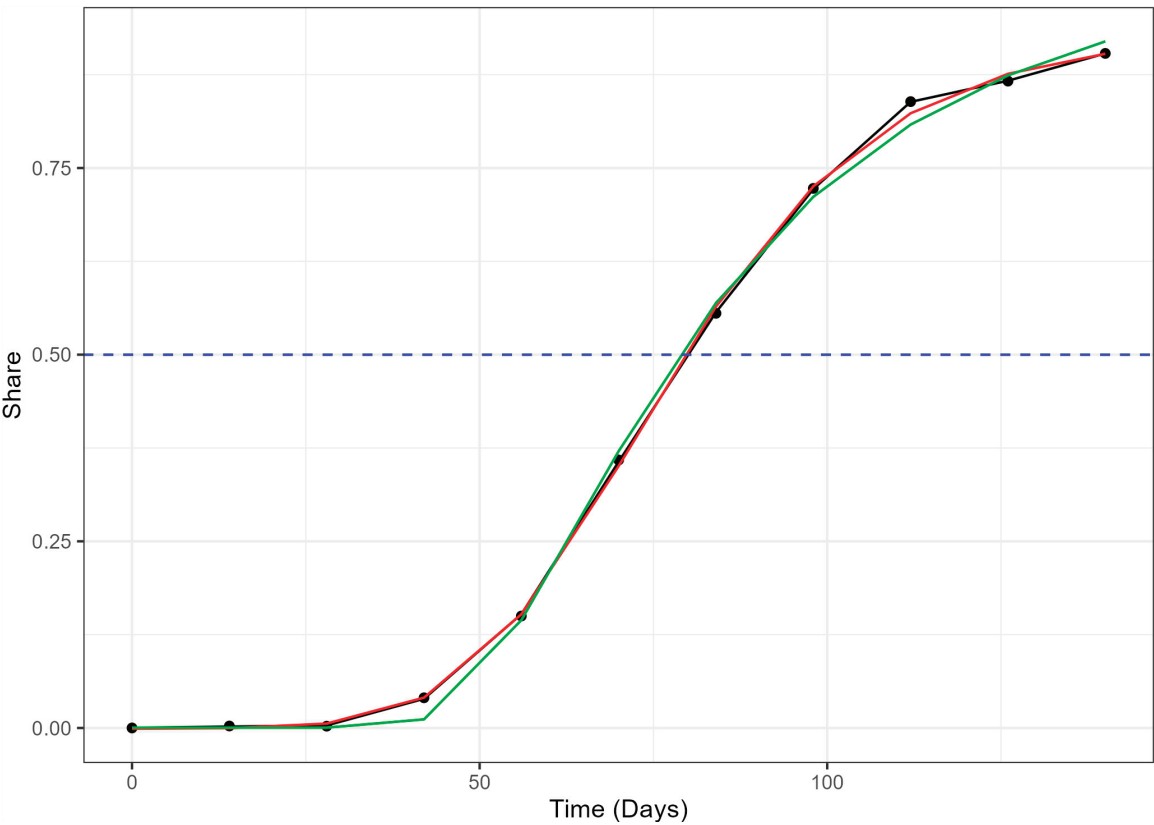

**Fig 10. The actual variant proportion data for XBB.1.5 in HHS Region 2, shown as black dots, plotted with the GAM (black line), logistic model (red), and Weibull model (green) fitted values.** The blue dashed line marks the 50% share level used for calculating the TTD values.

## Acknowledgments

The authors would like to acknowledge the University of Nevada, Las Vegas School of Public Health for their provided support in this study.

## Supporting information

**S1 File. Variant proportion data set.** This file includes the COVID-19 variant proportion at the time the secondary analysis was conducted.
(ZIP)

**S2 File. R Code and output.** This file includes the R code and output used for this study. The output was created using Quarto with a knitr engine.
(HTML)

## Author contributions

**Conceptualization:** Srishti Awasthi, Maryam Zolfaghari Dehkharghani, Miguel Fudolig.

**Data curation:** Srishti Awasthi, Miguel Fudolig.

**Formal analysis:** Miguel Fudolig.

**Investigation:** Srishti Awasthi, Maryam Zolfaghari Dehkharghani.

**Methodology:** Miguel Fudolig.

**Supervision:** Miguel Fudolig.

**Validation:** Miguel Fudolig.

**Visualization:** Miguel Fudolig.

**Writing original draft:** Srishti Awasthi, Maryam Zolfaghari Dehkharghani, Miguel Fudolig.

**Writing review & editing:** Srishti Awasthi, Maryam Zolfaghari Dehkharghani, Miguel Fudolig.

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
