## [Decision Letter · Decision Letter 0]

5 Nov 2024

PONE-D-24-41525

Emergence to dominance: Estimating time to dominance of SARS-CoV-2 variants using nonlinear statistical models

PLOS ONE

Dear Dr. Fudolig,

Thank you for submitting your manuscript to PLOS ONE. After careful consideration, we feel that your manuscript will likely be suitable for publication if it is revised to address the points below. Therefore, my decision is "Minor Revision." We invite you to submit a revised version of the manuscript that addresses the points raised during the review process. Please revise this paper.

We encourage you to submit your revision on Dec 20 2024 11:59PM. If you will need more time than this to complete your revisions, please reply to this message or contact the journal office at plosone@plos.org. Please include the following items when submitting your revised manuscript:

We look forward to receiving your revised manuscript.

Kind regards,

Oluwafemi Samson Balogun, Ph.D.

Academic Editor

PLOS ONE

Journal Requirements:

1. When submitting your revision, we need you to address these additional requirements. Please ensure that your manuscript meets PLOS ONE's style requirements, including those for file naming. The PLOS ONE style templates can be found at https://journals.plos.org/plosone/s/file?id=wjVg/PLOSOne_formatting_sample_main_body.pdf and https://journals.plos.org/plosone/s/file?id=ba62/PLOSOne_formatting_sample_title_authors_affiliations.pdf

Reviewers' comments:

Reviewer's Responses to Questions

**Comments to the Author**

1. Is the manuscript technically sound, and do the data support the conclusions?

Reviewer #1: Yes

2. Has the statistical analysis been performed appropriately and rigorously?

Reviewer #1: Yes

3. Have the authors made all data underlying the findings in their manuscript fully available?

Reviewer #1: Yes

4. Is the manuscript presented in an intelligible fashion and written in standard English?

Reviewer #1: Yes

5. Review Comments to the Author

Reviewer #1: The authors have tested the efficiency of Weibull models, logistic models and GAMs in the retrospective analysis of the emergence of the XBB.1.5 and JN.1 variants in the US. The accurate prediction of the fate of an emerging variant is important in the management of the public health response to pathogens, and this paper adds to that work. Here the focus is retrospective and phenomenological, attempting to see how well different non-mechanistic models were able to fit the curve after the fact, estimating both the shape of the curve and the time to dominance of the variant. The approach taken here seems reasonable, and the work answers the question asked.

An interesting question that could be asked with this data, that the authors do not attempt to address with this work, but I think would considerably add to the general interest of the manuscript is how well the different methods manage to estimate the time to dominance prior to this time, as this is issue that faces public health authorities looking forward in time attempting to plan a response. (Perhaps this could be done by adding data points in turn to see how the estimate improves with each extra days' worth of data).

Additionally, it would be nice if the authors could make the analysis code available as well as the data.

Beyond this I have some minor comments that should be addressed before publication:

Line 12-13: While it is true that the RNA polymerase is low fidelity, SARS-CoV-2 actually has a complicated proofreading mechanism to ensure genome stability, so this sentence should be amended. See: https://www.pnas.org/doi/pdf/10.1073/pnas.2106379119

Lines 15-21: This is a direct restatement of what is contained in lines 8-15. Did you meant to say this twice?

Lines 28-32: I would avoid saying that variants are "from" these countries. In some cases that will be true, but in others it may just represent where they were first detected, and it's impossible to tell which is which as the moment of emergence in unobserved.

Line 143: Is there a missing word at the end of this sentence?

Line 179: ED50 is not defined.

6. PLOS authors have the option to publish the peer review history of their article (what does this mean?). If published, this will include your full peer review and any attached files.

Reviewer #1: No

---

## [Author Response · Author response to Decision Letter 1]

11 Nov 2024

Please see attached file marked "Response to Reviewers" for formatted response.

Reviewer #1: The authors have tested the efficiency of Weibull models, logistic models and GAMs in the retrospective analysis of the emergence of the XBB.1.5 and JN.1 variants in the US. The accurate prediction of the fate of an emerging variant is important in the management of the public health response to pathogens, and this paper adds to that work. Here the focus is retrospective and phenomenological, attempting to see how well different non-mechanistic models were able to fit the curve after the fact, estimating both the shape of the curve and the time to dominance of the variant. The approach taken here seems reasonable, and the work answers the question asked.

An interesting question that could be asked with this data, that the authors do not attempt to address with this work, but I think would considerably add to the general interest of the manuscript is how well the different methods manage to estimate the time to dominance prior to this time, as this is issue that faces public health authorities looking forward in time attempting to plan a response. (Perhaps this could be done by adding data points in turn to see how the estimate improves with each extra days' worth of data).

This is a valid point to raise. Estimating the time to dominance prior to the time to dominance changes the focus from estimation to prediction. While we can use the models to extrapolate and predict the time to dominance that way, it could present some dangerous consequences if done blindly. For instance:

• There are VOIs that were expected to dominate in the population but were unable to do so because of the emergence of another highly infectious variant/subvariant. While the models can predict a time to dominance for these variants, the prediction might be incorrect due to reasons that cannot be accounted for using a phenomenological reason.

• To address your suggestion, we can add data points to improve TTD estimates. However, this is tough to expect from real data because of the limitations on the time it takes to perform genomic sequencing.

Both these problems could be circumvented by incorporating multi-strain simulations and testing the prediction and effect of lower temporal resolution between tests, but this is currently outside the scope of the paper. We included a detailed discussion of these points in Lines 289-303 and added a recommendation in our conclusions in Lines 365-366.

Additionally, it would be nice if the authors could make the analysis code available as well as the data.

Thank you for this suggestion. I have uploaded an HTML file of the R code and output. The data can be downloaded from the CDC website and I have also uploaded it as part of the supplementary material.

Beyond this I have some minor comments that should be addressed before publication:

Line 12-13: While it is true that the RNA polymerase is low fidelity, SARS-CoV-2 actually has a complicated proofreading mechanism to ensure genome stability, so this sentence should be amended. See: https://www.pnas.org/doi/pdf/10.1073/pnas.2106379119 Thank you for this comment. We have edited this to reflect the lack of proofreading mechanisms to ensure high fidelity (Line 13-14)

Lines 15-21: This is a direct restatement of what is contained in lines 8-15. Did you meant to say this twice? No. This was an error. We have removed Lines 15-21 in the edited manuscript. Thank you very much.

Lines 28-32: I would avoid saying that variants are "from" these countries. In some cases that will be true, but in others it may just represent where they were first detected, and it's impossible to tell which is which as the moment of emergence in unobserved. Thank you very much for this input. We recognize how this language could be harmful and misleading. We have edited “from” to “first detected in” and was applied to all listed variants. (Lines 24-27).

Line 143: Is there a missing word at the end of this sentence? Yes. It was a LaTeX call to the section “Estimating Time to Dominance” that yielded a blank because section numbers were deleted. The section name was included in the revision. Thank you for bringing this oversight to our attention. (Line 140)

Line 179: ED50 is not defined. Thank you for noticing this. We have included the definition of ED50. (Lines 176-178)

---

## [Decision Letter · Decision Letter 1]

12 Feb 2025

Emergence to dominance: Estimating time to dominance of SARS-CoV-2 variants using nonlinear statistical models

PONE-D-24-41525R1

Dear Dr. Fudolig,

We’re pleased to inform you that your manuscript has been judged scientifically suitable for publication and will be formally accepted for publication once it meets all outstanding technical requirements.

Kind regards,

Ranjan K. Mohapatra, PhD

Academic Editor

PLOS ONE

Additional Editor Comments (optional):

Reviewers' comments:

Reviewer's Responses to Questions

**Comments to the Author**

1. If the authors have adequately addressed your comments raised in a previous round of review and you feel that this manuscript is now acceptable for publication, you may indicate that here to bypass the “Comments to the Author” section, enter your conflict of interest statement in the “Confidential to Editor” section, and submit your "Accept" recommendation.

Reviewer #1: All comments have been addressed

Reviewer #2: All comments have been addressed

2. Is the manuscript technically sound, and do the data support the conclusions?

Reviewer #1: Yes

Reviewer #2: Yes

3. Has the statistical analysis been performed appropriately and rigorously?

Reviewer #1: Yes

Reviewer #2: Yes

4. Have the authors made all data underlying the findings in their manuscript fully available?

Reviewer #1: Yes

Reviewer #2: Yes

5. Is the manuscript presented in an intelligible fashion and written in standard English?

Reviewer #1: Yes

Reviewer #2: Yes

6. Review Comments to the Author

Reviewer #1: (No Response)

Reviewer #2: The author had improved the manuscript as per suggestions/comments. It is acceptable in current form.

7. PLOS authors have the option to publish the peer review history of their article (what does this mean?). If published, this will include your full peer review and any attached files.

Reviewer #1: No

Reviewer #2: No

---

## [Editor Report · Acceptance letter]

PONE-D-24-41525R1

PLOS ONE

Dear Dr. Fudolig,

I'm pleased to inform you that your manuscript has been deemed suitable for publication in PLOS ONE. Congratulations! Your manuscript is now being handed over to our production team.

Kind regards,

on behalf of

Dr. Ranjan K. Mohapatra

Academic Editor

PLOS ONE